# An Empirical Study of the Factors Influencing User Behavior of Fitness Software in College Students Based on UTAUT

**Chao Wang \*** , **Gencheng Wu, Xinyi Zhou and Yiman Lv**

School of Physical Education, Shaanxi Normal University, Xi'an 710119, China
\* Correspondence: wangchaotiyu@snnu.edu.cn; Tel.: +86-153-3910-3356

**Abstract:** Only one decade since the emergence of the first sports-related mobile app, although there is a large amount of fitness software, the quality is uneven, and some people still have concerns about whether to use fitness software. College students account for a large proportion of the number of people using fitness software; through empirical research on the factors affecting the use of fitness software, it is of great significance to further promote people's healthy behavior. This study investigates the factors that influence the user behavior of fitness software in college students and then addresses the promotion of better health behaviors among college students. Using a Likert scale, 994 college students (51.3% male, 48.7% female) in five universities (each university accounted for about 20% of students) were given questionnaires. Based on the Unified Theory of Acceptance and Use of Technology (UTAUT), assumptions were proposed, and a structural equation model (SEM) was constructed. The maximum likelihood method was used to analyze data and verify hypotheses. The results of the study show that social influence, performance expectancy, and effort expectancy significantly affect college students' behavioral intention. Behavioral intention and gender have a significant positive effect and a moderating effect on user behavior. The development of fitness software should consider the factors mentioned above to enhance the health levels of college students.

**Keywords:** UTAUT; college students; fitness software; user behavior; influential factors





## 1. Introduction

Appropriate physical exercise is recognized as an effective means to promote physical and mental health [1], reduce the risk of chronic diseases [2], and control weight [3]. However, the World Health Organization (WHO) has reported that one in four adults and four out of five adolescents do not exercise enough [4].

In China, although people have slightly enhanced their awareness of the advantages of physical exercise, many citizens suffer from chronic diseases and are in a sub-health state. High blood pressure and other cardiovascular diseases are becoming more prevalent among the younger generation [5].

Against this backdrop, the government attaches great importance to the population's health status. The State Council released several formal documents between July and September 2019 [5,6]. These documents reiterated that scientific physical activities could help significantly in disease prevention and promote citizens' health and well-being. They also encouraged the development of "Internet + sports" and e-commerce platforms to offer people convenient ways to exercise.

As a product of "Internet + sports," fitness software has gained popularity in China; it mainly guides users to engage in physical activity and improve their health level through electronic devices [7]. Several studies have illustrated the effectiveness of using fitness software to improve physical fitness [7,8]. However, some citizens hesitate to use it. Fitness software has not been around for long: the first sport-related mobile software only appeared in 2008 [9]. Thus, there is a long way to go for the development and qualitative improvement of fitness software.

College students are more interested in new things. The number of college students using fitness software is relatively large [10]. Therefore, it is representative to treat college students as respondents in investigating the factors affecting fitness software use. Previous studies have applied extended theoretical models [11–15]. However, the number of studies is small, and parts of their results are inconsistent. Thus, in this study, based on the Unified Theory of Acceptance and Use of Technology (UTAUT) [16], the main question we studied was the direction and extent to which various factors affected the behavior of college students using sports software. We also added two additional constructs into the model to explore the factors, namely perceived cost (PC) and perceived risk (PR). The significance of this study is that it can deepen the understanding of these influential factors, offer fitness software developers some scientific suggestions, and enable students to be more willing to use fitness software to improve their health conditions.

*Literature Review*

With the rapid development of modern society, people are no longer satisfied with the rapid enrichment of material life. This paper by Jian Wang et al. [17] analyzed the current situation of consumer demand for fitness and leisure activities in the context of big data, and it shows that consumer demand for fitness and leisure activities is expanding, that demand is diversifying and personalizing, and that the mobile internet has a significant impact on consumer demand for fitness and leisure activities. Several studies have demonstrated the effectiveness of consistent use of fitness software in helping people to improve their physical fitness. For example, Silke Neuman et al. [18] used interactive Exergame software to test activity-centered exercise training related to the daily tasks of older people in Switzerland, Spain and Germany over a period of three months with an average age of 70 years and showed improvements in physical function, daily activity performance and quality of life. CHING-TING HSU et al. [19] proposed and implemented a public fitness equipment IoT architecture, i.e., the interconnection of public fitness equipment hardware and software, providing users with exercise prescriptions on IoT devices using microprocessors and WiFi modules, and experimental results demonstrated significant improvements in physical fitness for all participants.

Parthasarathy et al. [20] show that it only takes about a fifth of the cost of acquiring a new user to maintain an existing user. Wang Zheng et al. [21] claim that user perception of value and satisfaction are the main factors influencing the continued use of the software, and that perceived value is also the main measure of user satisfaction. Meanwhile, the usefulness and fun of the software are the main considerations for users to make perceptual judgments. By surveying 324 fitness software users, Neeraj Dhiman et al. [22] found that important predictors of intention to adopt fitness software included expected effort, social influence, perceived value, habit and personal innovation. Personal innovation was the strongest predictor of behavioral intention, with factors such as performance expectations, convenience and hedonic motivation not influencing behavioral intention. Our study will build on previous work by surveying more fitness software users and considering gender differences.

## 2. Materials and Methods

### 2.1. Model Construction

Venkatesh et al. [16] brought together eight existing theories, including Theory of Reasoned Action (TRA), Theory of Planned Behavior (TPB), Technology Acceptance Model (TAM), Motivational Model (MM), Combined TAM and TPB (C-TAM-TPB), Model of PC Utilization (MPCU), Social Cognitive Theory (SCT), and Innovation Diffusion Theory (IDT), presented a new theory called UTAUT, and constructed a model based on it. This theory comprises two factors that work as direct determinants of use behavior (UB)—behavior intention (BI) and facilitating conditions (FC)—as well as three factors that have a determinant effect on BI: performance expectancy (PE), effort expectancy (EE), and social influence (SI). Moderating variables such as gender, age, experience, and voluntari-

ness of use are included in the model. The explanatory power of this model extends up to 70%, so this research was based on this model to investigate factors.

While the variance explained by the model is relatively high, additional research should try to add other constructs to offer an even richer understanding of technology adoption and use behavior [16]. Other studies also said that this model is imperfect and that researchers should add different variables to suit specific situations [21,22].

In studies associated with fitness software based on the theoretical model, Luo [11] extended TAM by adding perceived cost to explore influential factors of fitness software usage among college students; Hu et al. [12] extended UTAUT by adding perceived cost and risk and individual innovation to find factors that affect college students' intention to use health applications; Yuan et al. [13] applied UTAUT2, which differs from UTAUT in that it added hedonic motivation, price, and habit, to investigate the factors that affect intentions toward continued usage among college students. Herrmann et al. [14] applied TPB to study the factors influencing adults' fitness application usage and effectiveness. Liu et al. [15] applied UTAUT to investigate the factors influencing the intention to use fitness software among college students. Other studies on fitness-related software have consistently shown that users worry about inaccurate evaluations and information leakage when they use fitness software [12,23]. From what has been discussed, the present study aims to extend the UTAUT and focus on perceived cost (PC) and perceived risk (PR).

As fitness software has been around for about 12 years [9], the user experience remains insufficient. There is no noticeable difference in age among college students; this paper also aims to inspire active learning, which is considered an integral part of learning literacy and education [24,25]. Gender is an essential moderating variable in the Technology Acceptance Model and has been emphasized by several researchers [26,27]. Thus, this paper does not consider experience, age, and voluntariness of use, but gender is included. The final use behavior model of fitness software for college students was constructed (Figure 1) accordingly. The hypotheses are proposed as follows:

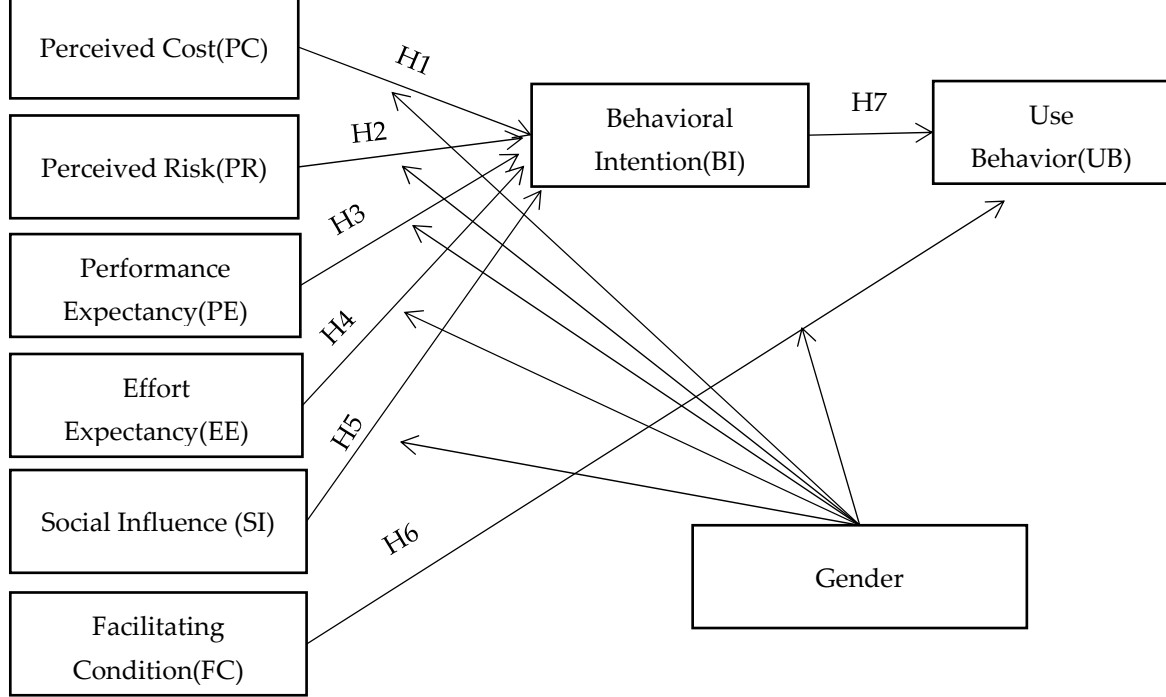

**Figure 1.** Use the behavior model of college students using fitness software.

**H1.** *PC has a negative effect on BI.*

Cost is what the pay users have to afford when using the software (i.e., the fee of online courses, necessary network traffic, etc.), so PC is defined as the degree to judge whether a college student believes the cost of using fitness software is high. The more money they need to pay, the less willing they have to adopt the software.

**H2.** *PR has a negative effect on BI.*

Risk is associated with the worries produced along with using the software. (i.e., information leakage, wrong guidance, etc.); thus, PR refers to the degree to which a university student believes that he or she will have to afford certain risks while using fitness software. More risks will prevent users' motivation to use it.

**H3.** *PE has a positive effect on BI.*

PE refers to the degree to which an individual thinks he or she can benefit from using a system and meet expectations [16]. In this study's context, college students can benefit from adopting the software (i.e., helping record their extent of exercise, controlling weight, socializing with others, etc.), strengthening their tendency to use it.

**H4.** *EE has a positive effect on BI.*

EE is defined as the degree of ease of using the system [16]. College students are more likely to find it simple to use the software due to their knowledge of electronic devices such as smartphones and computers [10], so they will be more willing to use it.

**H5.** *SI has a positive effect on BI.*

SI refers to the degree to which a person perceives the importance of others thinking he or she should use the system [16]. For college students, classmates, teachers, and parents are generally regarded as indispensable, and their supportive attitude toward the software will lead to students being more likely to use it.

**H6.** *FC has a positive effect on UB.*

FC is defined as the degree to which a person believes the infrastructure related to organization and technique supports the use of the system [16]. It is also explained as factors in the environment that impede or facilitate technology acceptance [28]. Thus, things such as suitable places for physical exercise and electronic equipment performance are included in FC. The more those objective conditions support the use of software, the more likely college students will use it.

**H7.** *BI has a positive effect on UB.*

### 2.2. Questionnaire Survey

The questionnaire comprised two parts: basic information, including gender, grade, and name of the university, and 24 questions designed to study eight constructs (PC, PR, PE, EE, SI, FC, BI, and UB) using a 5-point Likert scale.

The respondents were undergraduate and postgraduate students (non-sports majors) chosen from five universities in Xian, China, all of which do not force students to use fitness software. All respondents either used or did not use fitness software voluntarily.

During the pre-research, the researchers arrived at each school at the same time, briefly interviewed people on the sports field, distributed questionnaires within two hours, and agreed to take a repeat survey a week later. In total, 140 surveys were returned after screening to ensure an equal number of subjects and a 1:1 ratio of men to women in each school, and these questionnaires were tested for reliability, which showed that they all met the criteria for reliability and validity and fully supported a large-scale distribution at a later stage. After reliability and validity were tested, the questionnaire was slightly

adjusted. The formal field survey was conducted between 14 and 27 October 2020. Each questionnaire was anonymous and self-administered.

### 2.3. Statistical Analysis

Raw data were recorded in SPSS (SPSS Version 21.0, IBM Corporation, Armonk, NY, USA) Descriptive statistics were used to draw preliminary sample demographics and reliability analysis statistics. KMO and Bartlett's tests were applied to test the reliability and validity of the questionnaire.

Through first-order confirmatory factor analysis (CFA) processed by Amos (Amos Version 21.0, IBM Corporation, Armonk, NY, USA), the standardized factor loadings of each observed variable, R2, indicator error variances of observed variables, and correlation coefficients between constructs were obtained. According to the standardized factor loadings and indicator variances, the composite reliability (CR) and average variance extracted (AVE) of each construct were known. The discriminant validity between constructs was also known because of the square root of AVE and the correlation coefficients.

A structural equation model was constructed. The maximum likelihood method in Amos.23.0 was used to test the model fitness and verify the hypotheses.

## 3. Results

### 3.1. Sample Characteristics

The response rate was 98.7% (1066/1080), and the effective response rate was 93.2% (994/1066). As shown in Table 1, 510 were male (51.3%), and 484 were female (48.7%), indicating that the male-to-female ratio was almost balanced. An overwhelming majority were undergraduates, and postgraduates constituted a small proportion. The sample capacity in the five universities was approximately equal.

**Table 1.** Demographics of the sample.

|  | Options | Sample Capacity | Sample % |
| --- | --- | --- | --- |
| Gender | Male | 510 | 51.3 |
|  | Female | 484 | 48.7 |
| Grade | Freshman | 336 | 33.8 |
|  | Sophomore | 202 | 20.3 |
|  | Junior | 146 | 14.7 |
|  | Senior | 230 | 23.1 |
|  | Postgraduate | 80 | 8.0 |
| University | A | 206 | 20.7 |
|  | B | 200 | 20.1 |
|  | C | 190 | 19.1 |
|  | D | 196 | 19.7 |
|  | E | 202 | 20.3 |

Note A–E: the five universities that participated in this survey.

### 3.2. Results of Reliability and Validity

The reliability was tested through Cronbach's Alpha, R2, and CR. Cronbach's Alpha > 0.7 [29], several R2 > 0.5 in the model, and CR > 0.6 [30] show that the model is of good reliability. As listed in Table 2, the Cronbach's Alpha for each construct was more significant than 0.7, R2 was greater than 0.5 for most items, and the CR for each construct was more significant than 0.6. This means this model has good reliability.

Through KMO, factor loadings, and AVE, the validity was tested. KMO ranged between 0.7 and 0.8, indicating that the factors are suitable for analysis [31]. Factor loading ranged between 0.5 and 0.95, indicating good model fitness [30]. AVE > 0.5 indicated that the model had good convergent validity [32]. When the square root of the AVE of each construct is larger than the correlation coefficients between the build and others, these constructs have good discriminant validity [32,33]. As presented in Tables 2–4, the overall

KMO was 0.784 (sig. < 0.001), and every factor loading ranged between 0.5 and 0.95. All ages were more significant than 0.5, and the discriminant validity also met the demand. Thus, this model has good validity.

**Table 2.** Results of reliability and validity analysis.

| Construct | Item | Cronbach's Alpha | Factor Loading | R2 | CR | AVE |
|---|---|---|---|---|---|---|
| Use behavior (UB) | UB1<br>UB2 | 0.859 | 0.816<br>0.924 | 0.665<br>0.853 | 0.863 | 0.760 |
| Behavioral Intention (BI) | BI1<br>BI2<br>BI3 | 0.884 | 0.729<br>0.926<br>0.897 | 0.531<br>0.858<br>0.804 | 0.890 | 0.731 |
| Perceived Cost (PC) | PC1<br>PC2<br>PC3 | 0.761 | 0.764<br>0.774<br>0.629 | 0.584<br>0.599<br>0.396 | 0.768 | 0.526 |
| Perceived Risk (PR) | PR1<br>PR2<br>PR3 | 0.753 | 0.669<br>0.887<br>0.609 | 0.447<br>0.787<br>0.371 | 0.771 | 0.535 |
| Performance Expectancy (PE) | PE1<br>PE2<br>PE3<br>PE4 | 0.796 | 0.626<br>0.781<br>0.688<br>0.727 | 0.392<br>0.609<br>0.474<br>0.528 | 0.800 | 0.501 |
| Effort Expectancy (EE) | EE1<br>EE2<br>EE3 | 0.807 | 0.611<br>0.877<br>0.828 | 0.373<br>0.770<br>0.685 | 0.821 | 0.609 |
| Social Influence (SI) | SI1<br>SI2<br>SI3 | 0.775 | 0.678<br>0.806<br>0.724 | 0.460<br>0.650<br>0.525 | 0.781 | 0.546 |
| Facilitating Conditions (FC) | FC1<br>FC2<br>FC3 | 0.748 | 0.659<br>0.782<br>0.685 | 0.435<br>0.611<br>0.469 | 0.753 | 0.505 |

**Table 3.** KMO and Bartlett's test.

| Kaiser–Meyer–Olkin Measure of Sampling Adequacy | | 0.784 |
|---|---|---|
| Bartlett's test of sphericity | Approx. Chi-Square | 5010.400 |
| | df | 276 |
| | Sig. | 0.000 |

**Table 4.** Discriminant validity of constructs.

| | PC | PR | PE | EE | SI | FC | BI | UB |
|---|---|---|---|---|---|---|---|---|
| PC | 0.725 | | | | | | | |
| PR | 0.395 *** | 0.731 | | | | | | |
| PE | 0.111 | −0.082 | 0.708 | | | | | |
| EE | 0.031 | −0.090 | 0.364 *** | 0.780 | | | | |
| SI | 0.131 * | 0.037 | 0.399 *** | 0.220 *** | 0.739 | | | |
| FC | 0.084 | −0.016 | 0.420 *** | 0.439 *** | 0.353 *** | 0.711 | | |
| BI | 0.151 ** | 0.004 | 0.390 *** | 0.277 *** | 0.407 *** | 0.270 *** | 0.855 | |
| UB | 0.020 | 0.008 | 0.132 * | 0.138 ** | 0.328 *** | 0.205 *** | 0.376 *** | 0.872 |

Note: * $p < 0.05$; ** $p < 0.01$; *** $p < 0.001$.

### 3.3. Evaluation Results of the Structural Equation Model

Based on the fitness software user behavior model for college students, Amos.23.0 was used to construct the structural equation model. Before verifying the hypotheses, some goodness-of-fit indices had to be considered to evaluate the model fit. According to accommodations and practical applications of previous researchers [34–36], this study decided to apply the following indices: the ratio of x2 to its degree of freedom (x2/pdf) (x2 is called CMIN in the maximum likelihood method of Amos) [37], good-of-fit index (GFI), adjusted good-of-fit index (AGFI), normed fit index (NFI), comparative fit index (CFI), incremental fit index (IFI), Tucker–Lewis index (TLI) also called the non-normed fit index (NNFI), and the root mean square error of approximation (RMSEA).

After the evaluation of the initial model, the insignificant path coefficients of covariance (PR↔FC, PC↔FC, PR↔SI, PR↔EE, PC↔EE, PR↔PE, PC↔PE) were set to zero. The model was modified through suggested modification indices (MI). At the same time, the covariant relationships of e14↔e15, e8↔e9, and e21↔e22 were built one by one and did not break the rules of the structural equation model [37]. This was completed to ascertain a good fit between the model and the data.

As seen in Table 5, all indices were in the recommended range in evaluating the final structural equation model. Thus, the hypotheses can be verified as follows.

**Table 5.** Summary of fit indices for the final structural equation model nine.

| Fit Index | Recommended Value | Actual Value |
|---|---|---|
| CMIN/DF | <3.0 | 533.007/233 = 2.288 |
| GFI | >0.90 | 0.919 |
| AGFI | >0.90 | 0.902 |
| NFI | >0.90 | 0.901 |
| CFI | >0.90 | 0.938 |
| IFI | >0.90 | 0.938 |
| TLI(NNFI) | >0.90 | 0.926 |
| RMESA | <0.08 | 0.051 |

Note: CMIN/DF: Chi-Square/Degree of Freedom; GFI: Goodness of Fit Index; AGFI: Adjusted Goodness of Fit; NFI: Normal Fit Index; CFI: Comparative Fit Index; IFI: Incremental Fit Index; TLI(NNFI): Tucker-Lewis Index (Non-Normed Fit Index); RMESA: Root Mean Square Error of Approximation.

### 3.4. Results of Hypotheses Testing

As shown in Table 6, except for H1, H2, and H6, the results of other path coefficients demonstrated that all the other four hypotheses were supported.

**Table 6.** Hypothesis testing results.

| Hypothesis | Path | Path Coefficient | Results |
|---|---|---|---|
| H1 | PC→BI | 0.092 | Not supported |
| H2 | PR→BI | −0.010 | Not supported |
| H3 | PE→BI | 0.256 *** | Supported |
| H4 | EE→BI | 0.107 * | Supported |
| H5 | SI→BI | 0.286 *** | Supported |
| H6 | FC→UB | 0.099 | Not supported |
| H7 | BI→UB | 0.350 *** | Supported |

Note: * $p < 0.05$; *** $p < 0.001$.

After the gender moderating variable was added, there was an apparent difference between male and female students. As seen in Table 7, H4, H5, H6, and H7 were supported for males, whereas H3, H5, and H7 were supported for females, and other hypotheses were not supported. Thus, gender has a moderating effect on UB.

**Table 7.** Testing results of the gender moderating effect.

| Hypothesis | Path | Male Path Coefficient | Male Results | Female Path Coefficient | Female Results |
|---|---|---|---|---|---|
| H1 | PC→BI | 0.088 | Not supported | 0.139 | Not supported |
| H2 | PR→BI | 0.003 | Not supported | −0.028 | Not supported |
| H3 | PE→BI | 0.140 | Not supported | 0.376 *** | Supported |
| H4 | EE→BI | 0.155 * | supported | 0.053 | Not supported |
| H5 | SI→BI | 0.252 ** | supported | 0.316 *** | Supported |
| H6 | FC→UB | 0.148 * | supported | −0.037 | Not supported |
| H7 | BI→UB | 0.323 *** | supported | 0.388 *** | Supported |

Note: * $p < 0.05$; ** $p < 0.01$; *** $p < 0.001$.

## 4. Discussion

Based on UTAUT, we sent questionnaires and constructed a structural equation model to investigate the factors that influence fitness software user behavior in college students. According to the results and the absolute value of the path coefficient, SI is the most vital factor with a significant positive effect on BI, which is consistent with previous research [15] but not with the other two studies [13,14]. The divergence is probably because of the time of investigation, the respondents, and the areas. The two essays were published in 2015 and 2017, respectively (the actual investigating time would have been earlier). At that time, fitness software was different from what it is today. In terms of respondents and areas, the former chose 64 adults aged 18 years and above in a Midwest suburban fitness center in the United States. The latter chose 317 students from a Midwestern university in the United States, in which 74% were white and the male-to-female ratio was not balanced. From the perspective of conformity in social psychology, a group can influence an individual to behave in the same manner as the others in the group [38], so it is reasonable for college students to be affected by others when they perceive people around them using fitness software. This can increase their intention to use such software. Therefore, software developers should take SI into account. They can improve online social circles, encourage users to share fitness information on other social media platforms, and cooperate with other popular online We-Media. There would be more people engaging in using the software. Administrators of universities can require professors and lecturers to use such software to impact their students positively.

PE is the second most vital factor having a significant positive effect on BI, which is consistent with all previous studies [11,13,23]. Therefore, developers should diversify the functions and guarantee the quality of the software. In addition to normal sports activities such as walking and running, some sports need to be considered, such as climbing, cycling, yoga, etc. The online to offline (O2O) transition can complement the software's function. For instance, video-based classes such as soccer, dance, health, and diet can be held online, and users can acquire skills through the software and practice offline.

EE also plays an important role and has a significant positive effect on BI, which is consistent with previous studies [11,15]. From the perspective of motivation, while playing a sport involves great difficulty and people have to put in a lot of effort and time to grasp the skills, motivation is often inhibited [39]. Therefore, developers should try to make it easier for users to use their software. The operation interface should be straightforward. Explanations of the courses should be mixed with demonstrations. The content of the systems themselves should be divided into different levels.

With three determinant factors positively influencing BI, BI has a significant impact on UB, which is consistent with all previous studies on technology acceptance.

However, FC does not have a significant effect on BI. This is consistent with one previous study [13]. This is possible because of the rapid development of the Internet and electronic devices in China. An overwhelming majority of college students use advanced equipment, so they may not be sensitive to the convenience that those devices have ushered

in. Another reason may be the nature of fitness software, which is less restricted by place or sports field [40]. Further studies are needed to confirm this finding.

PC and PR are not significant for BI. The results for PC are not consistent with previous studies [11–13]. This is possible because of the differences in the time of investigation and the respondent's location. The result of PR is not compatible with previous research [12], which is partly because there is a slight difference in the study area. The health applications they mentioned included both fitness and electronic medicine applications, but we only focus on fitness software. Therefore, whether PC and PR are significant predictors of BI remains disputable. Based on Chinese specific national conditions, China is in a decisive stage for building a moderately prosperous society in all respects. This means that the family backgrounds of college students are increasingly well-off. Thus, the sensitivity of price may correspondingly decrease. There is a general concern that college students are less aware of risk prevention and may not know how to deal with risk using software [41,42]. Overall, we believe that it may be reasonable that PC and PR do not significantly affect BI and that the results are of the reference value.

After adding gender as a moderating variable, there was a noticeable difference between male and female students. Among male students, SI and EE had a significant positive effect on BI, while BI and FC had a significant positive impact on UB. SI and PE significantly affect BI among female students, while BI has a significant positive effect on UB. Other path coefficients seen in this study are not effective. Although gender is a crucial variable [26,27], no previous research has included gender as a moderating variable in the model-based fitness software study area. Thus, we were not able to compare the results. However, one study shows that male college students' degree in self-assurance and self-evaluation is higher than that of female students [43], which may have caused the difference. However, the specific reason for the results remains unknown, and future studies should investigate the differences based on gender in fitness software use.

## 5. Conclusions

This study, based on UTAUT, focuses on investigating the factors that can influence the UB of college students using fitness software. The four hypotheses H3, H4, H5 and H7 were verified by data, and H1, H2 and H6 were not strongly supported by the analysis results. Our findings show that SI, PE, and EE have significant positive effects on college students' BI using fitness software. BI has a significant positive impact on UB, and gender has a moderating effect. In the future, the development of fitness software for college students should be considered from the perspectives of exerting SI improve, the online social circle, commenting on the function of fitness software, improving the reward mechanism of users sharing fitness information to other social media, expanding the influence of fitness software, and then participating in fitness. It is also important to enrich the function of the fitness software itself. According to the principle of more diversified fitness methods, they should not be limited to walking, running, aerobics, etc. but rather should be combined with the current popular sports, such as mountaineering, cycling, yoga, etc.. It could also be more integrated with O2O (Online to Offline) mode and could add all kinds of teaching videos, such as football, basketball, volleyball, dancesport, table tennis, healthy diet, etc., for users to learn and practice online. Reducing the difficulty of using the software, the operation interface should be simple and easy to understand, the steps of the navigation key should be clear, and the course content should be difficult and easy to stratify to meet the needs of different users. Focusing on gender preference to increase the health levels of college students, it could also be beneficial to add the category recommendation column according to the user's gender at the time of registration. Male students prefer ease of use and push content from the perspective of simple interface and strong practicality. Girls tend to prefer software functions and push content as rich as possible in project selection and course types.

The survey objects in this paper are college students in the same area. The results and data obtained may have regional characteristics and cannot represent the usage of

all college students, so the universality needs to be considered. When designing fitness apps for college students, attention should be paid to the entertainment and sociability of fitness apps.

By studying user behavior and discussing the particularity of different behavior-influencing factors, our studies enrich the existing system of fitness software users. The results will help fitness software improvements and upgrades, provide users with targeted services, enhance the user experience degrees, attract new users, increase the rate of user retention, promote people to form a healthy exercise habit and then establish a sustainable healthy life pattern, and continuously improve people's physical health level.

Due to the short history of fitness software, this paper only investigated college students without considering the influence of age, experience, and other variables. The future research can expand the research scope, level and so on with more in-depth research. We will optimize and supplement the existing model. In our future research and analysis, more variables will be added to explore the influence of various factors on the behavior of college students using fitness software.

**Author Contributions:** Conceptualization, C.W.; methodology, C.W. and G.W.; software, X.Z.; validation, X.Z., G.W. and Y.L.; formal analysis, X.Z. and G.W.; investigation, C.W.; resources, Y.L.; data curation, X.Z.; writing—original draft preparation, C.W.; writing—review and editing, G.W.; visualization, X.Z.; supervision, C.W.; project administration, C.W.; funding acquisition, Y.L. All authors have read and agreed to the published version of the manuscript.

**Funding:** This research was funded by the Social Science Support Fund of Shaanxi Normal University, grant number No. 2017.

**Institutional Review Board Statement:** The study was conducted in accordance with the Declaration of Helsinki, and approved by the Ethics Committee of Shaanxi Normal University (202016001, 20 May 2020.)

**Informed Consent Statement:** Informed consent was obtained from all subjects involved in the study. Written informed consent has been obtained from the patient(s) to publish this paper.

**Data Availability Statement:** Data available on request due to restrictions on privacy, the data presented in this study are available on request from the corresponding author. The data are not publicly available due to privacy restrictions. The corresponding author can be contacted by email to request data sharing.

**Conflicts of Interest:** The authors declare no conflict of interest. The funders had no role in the study's design, in the collection, analyses, or interpretation of data, in the writing of the manuscript, or in the decision to publish the results.

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
