# Peer review of "An Empirical Study of the Factors Influencing User Behavior of Fitness Software in College Students Based on UTAUT"

_sustainability, doi:10.3390/su14159720_

Round 1

Reviewer 1 Report

Dear Author(s),

Thank you for your paper. I would like to underline that I found the topic interesting and with several aspects and elements of novelty.  Your paper is submitted in the section “Health, Well-Being and Sustainability.

However, a connection with “Sustainability” concept could be clarified better. Your point of view about it, is not very clear.

Could be help relevant to add recent publications to your References and clarify following points:

·         Why you´ve chosen Venkateshs´ Model Construction?

·         What kinf of adjustments were made in questionnaire in the pre-research stage?

·         What is a connection with sustainability?

·         How questionnaires were administrated? Paper? On-line? How did you select participants (sample)?

Thank you again for your work and I wish you all the best in your future research!

Author Response

Dear Reviewers,

        Thank you for your review, below is my answer to your question, the revised paper is in the attachment.

1.Why you´ve chosen Venkateshs´ Model Construction?

    Integrated technology acceptance model (UTAUT) introduces community factors and convenience conditions on the basis of technology acceptance model (TAM), and becomes a more comprehensive theoretical model to study user behavior. The UTAUT model integrates the eight it acceptance variables of the technology acceptance model into four core variables, namely, performance expectation, effort expectation, community influence and contributing factors. The model emphasizes that community factors are the direct factors that affect users' behavior, and people who are important to users will have a direct influence on individual behavior. In addition, convenience conditions, that is, objective environmental factors will also affect the user's difficulty in using the system. The integrated technology acceptance model explains the acceptance of information technology more strongly than the previous model.

2.What kinf of adjustments were made in questionnaire in the pre-research stage?

    In the pre-survey stage, the team made a comprehensive analysis of the recovered questionnaires and found that all the sports and fitness software used by the subjects were within the scope of our consideration. The following two survey questions were unnecessary, so they were removed from the formal questionnaire:
(1) Do you use fitness software?â–¡Yes    â–¡no
(2)What kind of fitness software do you use? â–¡Yue moving circle and other data   â–¡Keep and other tutorial classes   â–¡Wechat sports, QQ sports and other social classes  â–¡Alipay steps and other public welfare categoriesâ–¡Ham sports and other educational categories   â–¡other

3.What is a connection with sustainability?

    By studying user behavior and user behavior, the continuous use to discuss the particularity of different behavior influence factor, to enrich the existing system of fitness software users use the behavior theory research, the results will help fitness software improvements and upgrades, to provide users with targeted services, enhance the user experience degrees, attract new users, increase the rate of user retention, Promote people to form a healthy exercise habit, and then establish a sustainable healthy life pattern, and continuously improve people's physical health level.

4.How questionnaires were administrated? Paper? On-line? How did you select participants (sample)?

    The paper version of the questionnaire was distributed at five colleges that did not mandate the use of fitness software. 200 questionnaires were distributed to each school, with a 1:1 male to female ratio, randomly distributed on the sports field and accompanied by interviews.

Reviewer 2 Report

Review report for the paper "An Empirical Study of the Factors Influencing User Behavior of Fitness Software in College Students based on UTAUT":

The main remarks are the following:

1. Please avoid using the number of hypotheses (H3, H4, ..., H7) in the Abstract. Try to use a "normal" language for the reader. The abstract must shortly present the goal and the main findings in a regular language.

2. In the Introduction, please present and describe the research question. You described the research gap and the research goal, but I didn't find the research question. Also, you should shortly present the main chapters of your article.

3. Within Figure 1 (rows 74-75), draw the hypotheses (H1, H2, ..., H7) so that the readers have a graphical representation of the research hypotheses. Add the hypotheses numbers on the arrows.

4. The list of references should be improved in order to complete the context of your research. I recommend you to cite in your article the following articles: https://doi.org/10.1089/g4h.2017.0079, https://doi.org/10.37394/232018.2020.8.3, https://doi.org/10.1007/s10586-018-2207-y, https://doi.org/10.24818/ie2020.02.01, https://doi.org/10.1108/JIBR-05-2018-0158. Maybe a distinct section named "Literature Review" would be useful for the readers.

5. In the "2.1. Model Construction" section, you describe the hypotheses. In my opinion, you should insist on the arguments for H1, H2, H3, H4 by citing additional resources from literature. Now it seems that these hypotheses are based only on your own feelings. Try to impove this aspect.

6. In "2.2. Questionnaire Survey" you tell that you used 24 questions designed to study 8 constructs. I recommend you to include an appendix at the end of the article and present there the questions. This way, the readers will fully understand your survey.

7. The Discussion section is well written and it doesn't need any improvement.

8. In the Conclusion section, please present the following aspects:

- the research limitations;

- the managerial implications;

- the future research directions.

Kind Regards!

Author Response

Dear Reviewer,

    Thank you for your review, below is my answer to your question, the revised paper is in the attachment.

Question1:It has been modified as you suggested.

Question2:It has been modified as you suggested.

Question3:It has been modified as you suggested.

Question4:Thank you for your references, which have been refined in accordance with your suggestions.

Question5:Thanks for your suggestion, the references that support our hypothesis have been briefly stated above.

Question6:The questionnaire is attached.

Question7:Thank you for your affirmation.

Question8:It has been modified as you suggested.

Reviewer 3 Report

The topic addressed an interesting topic. In fact, the paper has a clear interest from different scientific disciplines and therefore it could have a remarkable impact in the specialized scientific literature.

In my view, the paper is hardly related to the journal's editorial line, addresses a topic of particular interest that may have a clear interest for readers, makes a significant contribution, and opens a new research line that may lead to future works in the same direction.

Considering the subject's interest, I would like to make some comments and suggestions that should always be understood positively and considering that the different observations constitute different avenues that may improve this exciting research and facilitate its publication and impact in the subsequent specialized literature.

Abstract: I would complete the abstract with a general idea and on the approached theory (specialized literature). The abstract begins abruptly with information about the methodology used, but without presenting the context and need for such research.

After the Introduction section go directly to the 2. Materials and Methods section. The Literature Review section: The scientific context of the approached topic is missing.

2.2. Questionnaire Survey: A total of 140 questionnaires were sent out in the pre-research stage. What was the sampling method used? What was the formula used to determine the sample size, which would make the research representative? How were the participants selected? Why is the number of respondents considered sufficient to interpret the results and generalize them? What is the entire statistical population?

Conclusion: Please elaborate on the conclusion and explain your findings better. Please, extend the specific results to their broader implications, which can then be tied in with the general background given in the introduction to maximize the impact of the overall paper. Start by stating whether your hypothesis was supported. Interpret the results: what do the results imply? (Do not simply repeat the results again and do not make interpretations that are not supported by the data).

The limits of the research are missing. Any research has its own limits generated by the tools used, the methodology used, the territorial area approached or even the topic itself.

The implications section is missing: What are the implications of this article in the literature? Who does these results help (both scientifically and professionally)? Please, improve the explanations of what we have learned, develop and extend the managerial implications

Based on these comments, my position is to revise the manuscript in-depth before resubmitting the paper again. However, I hope that all these comments will serve the author to improve the quality of the paper.

Author Response

Dear Reviewer,

    Thank you for your review, below is my answer to your question, the revised paper is in the attachment.

Question1:Thanks for your suggestion.It has been modified as you suggested.

Question2:First of all, this research method meets the requirements of this field. After the sample size is determined, a preliminary survey is conducted in accordance with the survey practice to determine the reliability and validity of the questionnaire. Before the formal investigation, we have conducted a preliminary screening of the subject group and determined the sample size of the study to be about 1000 copies. It is customary for a first round of random samples selected, and then according to the time and space, equal the number of all the school subjects, factors such as sex ratio 1:1, eventually determine 28 a preliminary investigation at each school, a total of 140, and 140 questionnaires test, reliability and validity of the results of its meet the standard, the reliability and validity of late completely support area.

Question3:Thanks for your suggestion.It has been modified as you suggested.

Question4:The survey objects in this paper are college students in the same area. The results and data obtained may have regional characteristics and cannot represent the usage of all college students, so the universality needs to be considered. When designing fitness apps for college students, attention should be paid to the entertainment and sociability of fitness apps.

Question5:By studying user behavior and user behavior, the continuous use to discuss the particularity of different behavior influence factor, to enrich the existing system of fitness software users use the behavior theory research, the results will help fitness software improvements and upgrades, to provide users with targeted services, enhance the user experience degrees, attract new users, increase the rate of user retention, Promote people to form a healthy exercise habit, and then establish a sustainable healthy life pattern, and continuously improve people's physical health level.

Round 2

Reviewer 2 Report

Dear authors,

I appreciate the fact that you have improved the structure and quality of the article.

Good luck!

Author Response

Dear reviewer:

       Thank you for your help and recognition.

Reviewer 3 Report

The article shows improvements over the original version, but some of the information requested in the first review is still missing. 

For the article to be suitable for publication, it should (in my opinion) be improved on the following aspects:

- literature review is missing (it was a request since the first review). The literature review section creates the context of the topic by presenting data and information including references to other similar research.  (Suggestion: considering that the literature is not very rich in similar content, I would recommend that some of the information presented in the introduction be put in a separate section called "literature review". In order not to keep the introduction too short, I would recommend a free wording, a narrative of the research idea (without the obligatory presence of citations), so as not to repeat the same citations, the same text both in the introduction and in the literature review section.)

- Regarding the justification for the number of questionnaires used, the authors have provided a suitable explanation in the response form. I would suggest that this explanation be included in the text of the article. ("First of all, this research method meets the requirements of this field. After the sample size is determined, a preliminary survey is conducted in accordance with the survey practice to determine the reliability and validity of the questionnaire. Before the formal investigation, we have conducted a preliminary screening of the subject group and determined the sample size of the study to be about 1000 copies. It is customary for a first round of random samples selected, and then according to the time and space, equal the number of all the school subjects, factors such as sex ratio 1:1, eventually determine 28 a preliminary investigation at each school, a total of 140, and 140 questionnaires test, reliability and validity of the results of its meet the standard, the reliability and validity of late completely support area"

Author Response

Respected reviewer:

        Thank you for your suggestion, we have made the changes as you suggested and the revised article is attached.
